# Analogies and Relations between Non-Additive Entropy Formulas and Gintropy

**DOI:** 10.3390/e26030185

**Published:** 2024-02-22

**Authors:** Tamás S. Biró, András Telcs, Antal Jakovác

**Affiliations:** 1HUN-REN Wigner Research Centre for Physics, 1121 Budapest, Hungary; telcs.andras@wigner.hun-ren.hu (A.T.); jakovac.antal@wigner.hun-ren.hu (A.J.); 2Hungarian Physics Department, Physics Faculty, University Babeş-Bolyai, 400084 Cluj-Napoca, Romania; 3Complexity Science Hub, 1080 Vienna, Austria

**Keywords:** entropy, Gini index, Lorenz curve, non-extensive

## Abstract

We explore formal similarities and mathematical transformation formulas between general trace-form entropies and the Gini index, originally used in quantifying income and wealth inequalities. We utilize the notion of gintropy introduced in our earlier works as a certain property of the Lorenz curve drawn in the map of the tail-integrated cumulative population and wealth fractions. In particular, we rediscover Tsallis’ *q*-entropy formula related to the Pareto distribution. As a novel result, we express the traditional entropy in terms of gintropy and reconstruct further non-additive formulas. A dynamical model calculation of the evolution of Gini index is also presented.

## 1. Motivation

This paper responds to a call by the journal Entropy to accompany various contributions in honor of Constantino Tsallis’ 80th birthday. Professor Tsallis initiated the field of non-extensive statistical mechanics with his seminal paper in 1988 [1] and kept this field flourishing with his continuous activity since then. One of his recent books on Non-Extensive Statistical Mechanics [2], has the subtitle “Approaching a Complex World”. It characterizes the range of research fields, beyond physics, where non-additive entropy formulas can be applied [3,4,5,6,7]. Adding a physicist’s approach to the mathematical predecessor formulas, such as Rényi entropy [8], and further generalizations of the Boltzmannian log-formula proliferating in the field of informatics and mathematics [9,10,11,12], his work is acknowledged to date in a wide and strengthening community of researchers dealing with complexity [13,14,15,16,17,18,19,20].

Over the years, newcomers and opponents of non-extensive thermodynamics have often argued that using any formula between entropy and probability besides the classical Boltzmann–Gibbs–Shannon version can only then be generally applied, and it is advised to use it if it moves beyond merely being an alternative formal possibility—when it must be applied. Therefore, there is an ongoing challenge to find real-world data and applications that can only be described by a non-Boltzmannian entropy formula. Such cases are found with increasing frequency in complex systems. An interesting approach is presented in [21]: it shows how to analyze nuclear production data to reveal non-extensive thermodynamics. (Our earlier calculations of fluctuations and deviations from an exponential kinetic energy distribution due to the finiteness of a heat bath, presented in several publications, should not be cited here, because the Editors at MDPI consider self-citations, even one sixth of the total, to be biased and unnecessary.)

The Tsallis and Rényi entropy formulas are monotonic functions of one another; therefore, their respective canonical equilibrium distribution functions coincide, not accounting for constant factors related to the partition sum. Since the Rényi entropy is defined as
(1)$$S_{R}\;=\;\frac{1}{1-q}\ln\sum_{i}p_{i}^{q},$$
and the Tsallis entropy as
(2)$$S_{T}\;=\;\frac{\sum_{i}\left(p_{i}^{q}-p_{i}\right)}{1-q}\;=\;\frac{e^{S_{R}\left(1-q\right)}-1}{1-q},$$
one obtains, in the canonical approach to the physical energy distribution,
(3)$$\frac{\partial S_{T}}{\partial p_{i}}\;=\;e^{S_{R}\left(1-q\right)}\;\frac{\partial S_{R}}{\partial p_{i}}\;=\;\beta E_{i}+\alpha .$$The actual energy level is denoted by $E_{i}$ in this formula, while α and β are Lagrange multipliers. The former is related to the partition function and the latter to the absolute temperature (via the average value of the energy). The prefactor Equation (Equation 3) is independent of $p_{i}$; therefore, the functional forms of the canonical PDFs coincide, reconstructing the Pareto or Lomax distribution [22,23,24,25].

In a microcanonical approach, all trace-form entropies are maximal at the distribution uniform in *x*, provided that the non-trivial function in the formula satisfies the general properties of non-negativity and convexity. Constraining the expectation value of the base variable, $\langle x\rangle$, of which we intend to study the probability density function, P(x), leads to an entropy depending on the constrained value, say $\alpha +\beta \langle E\rangle$ for an energy (*E*) distribution. These functions, of course, vary. The properties of entropy formulas also differ: while the Rényi entropy is additive for the factorization of probabilities and the Tsallis *q*-entropy is not, the *q*-entropy is formally an expectation value and the Rényi entropy is not.

In this paper, we first briefly review the Gini index and the Lorenz curve, spanning a map of the tail-cumulative fractions of a population and the wealth owned by this population. We furthermore review the definition and basic properties of gintropy, defined as the difference between the above two cumulatives. Following this, we introduce some gintropy formulas being formal doubles of well-known and used entropies. Finally, we explore the transformations from one (entropic) view to the other (gintropic view) and present a dynamical model calculation of the evolution of the Gini index based on a master equation.

## 2. About Gintropy

In our search for additional motivation for the use of non-Boltzmannian entropy formulas, we encounter the Gini index [26,27,28], classically used in income and wealth data analyses. It measures the expectation value of the absolute difference, $\langle \left|x-y\right|\rangle$, normalized by that of the sum, $\langle x+y\rangle =2\langle x\rangle$, when taking both variables from the same distribution. It delivers values between zero and one (100%):(4)$$G\;=\;\frac{\langle \left|x-y\right|\rangle }{\langle x+y\rangle }\;=\;\frac{1}{2\langle x\rangle }\;\int_{0}^{\infty }\;dx\int_{x}^{\infty }\;dy\left|y-x\right|P\left(x\right)P\left(y\right).$$Here, P(x) is the underlying PDF. This formula can be transformed into several alternate forms, as has been shown in Ref. [29] in detail. We have also found that a function defined by tail-cumulative functions, gintropy, has properties very similar to those of an entropy–probability trace formula function.

Two basic tail-cumulative functions constitute the definition and usefulness of gintropy. The first is the cumulative population,
(5)$$\bar{C}\left(x\right)\;\equiv \;\int_{x}^{\infty }\;dy\;P\left(y\right),$$
and the second is the cumulative wealth normalized by its average value (also called the scaled and (from below) truncated expectation value),
(6)$$\bar{F}\left(x\right)\;\equiv \;\int_{x}^{\infty }\;dy\;\frac{y}{\langle x\rangle }P\left(y\right).$$We note here that the notions “population” and “wealth” are used in a general sense: any type of real random variable *x* associated with a well-defined PDF, P(x), has a tail-cumulative fraction (cf. Equation (Equation 5)) and a scaled fraction of the occurrence of the basic variable defined in Equation (Equation 6). For example, *x* may denote the number of citations that an individual author receives and P(x) the distribution of this number in the analyzed population. Then, $\bar{C}\left(x\right)$ is the fraction of papers cited *x* times or more, and $\bar{F}\left(x\right)$ is the fraction of citations received for these relative to all citations [30]. The above definitions and the following analysis of gintropy can be used for any PDF defined on non-negative variables x≥0 and having a finite expectation value.

The Lorenz map [31] plots the essence of a PDF on a $\bar{C}-\bar{F}$ plane. Since always $\bar{F}\geq \bar{C}$, following from the positivity of the PDF, P(x), the Lorenz curve always runs on this map above or on the diagonal. At x=∞, both quantities are vanishing, $\bar{F}\left(\infty \right)=\bar{C}\left(\infty \right)=0$, because the integration range shrinks to zero, and they also coincide at x=0, following from their normalized definitions: $\bar{F}\left(0\right)=\bar{C}\left(0\right)=1$. The Gini index can be described as the area fraction between the Lorenz curve and the diagonal to the whole upper triangle (with an area of 1/2). The quantity of gintropy, introduced by us in an earlier work [29], is the difference
(7)$$\sigma \;\equiv \;\bar{F}-\bar{C}.$$This is a function of the fiducial variable *x*, and it vanishes as a function only for those PDFs that allow only a single value for *x*. The gintropy is non-negative and it shows a definite sign of curvature. On the Lorenz map, it is best viewed and expressed as a function of $\bar{C}$. The connection between these two variables, derived from Equation (Equation 5), is given by $d\bar{C}/dx=-P\left(x\right)$. Likewise, $d\bar{F}/dx=-xP\left(x\right)/\langle x\rangle$ follows from the definition in Equation (Equation 6). Then, it is easy to establish that it has a maximum exactly at the average case, $x=\langle x\rangle$:(8)$$\frac{d\sigma }{d\bar{C}}\;=\;\frac{d\bar{F}}{d\bar{C}}-1\;=\;\frac{x}{\langle x\rangle }-1.$$The second derivative of gintropy in the Lorenz map is always negative:(9)$$\frac{d^{2}\sigma }{d{\bar{C}}^{2}}\;=\;\frac{1}{\langle x\rangle }\;\frac{dx}{d\bar{C}}\;=\;-\frac{1}{\langle x\rangle P\left(x\right)}\;<\;0.$$As a consequence, the gintropy, $\sigma (\bar{C})$, has a single maximum (between two maxima, there would be a region with an opposite-sign second derivative for a continuous function). This maximum can be expressed as a function of the average value:(10)$$\sigma_{\max}\;=\;\bar{F}\left(\langle x\rangle \right)-\bar{C}\left(\langle x\rangle \right).$$Finally, the Gini index itself is twice the area under the gintropy:(11)$$G\;=\;2\int_{0}^{1}\;d\bar{C}\;\sigma \left(\bar{C}\right).$$

## 3. Entropy from Gintropy

It is important to consider a few simple cases for gintropy. First of all, a PDF allowing only a singular value, such as P(x)=δ(x−a), leads to vanishing gintropy. Then, σ=0 for all $\bar{C}\in \left[0,1\right]$. This case is degenerate; the second derivative is also zero across the whole interval and there is no definite maximum. A few examples have been discussed in Ref. [29]. Here, we use the Tsallis–Pareto distribution, as a limiting case, as it includes the Boltzmann–Gibbs exponential too. The tail-cumulative function is given as a two-parameter set with a power-law tail and the proper $\bar{C}\left(0\right)=1$ normalization:(12)$$\bar{C}\left(x\right)\;=\;{\left(1+ax\right)}^{-b}.$$Here, *a* and *b* are positive. It follows a PDF,
(13)$$P\left(x\right)\;=\;ab{\left(1+ax\right)}^{-b-1},$$
an expectation value of $\langle x\rangle =1/a\left(b-1\right)$, and finally a gintropy formula:(14)$$\sigma \;=\;abx{\left(1+ax\right)}^{-b}\;=\;b\left({\bar{C}}^{1-1/b}\;-\;\bar{C}\right).$$Related to the more popular form, one uses q=1−1/b as a parameter and arrives at the q-gintropy formula:(15)$$\sigma_{q}\left(\bar{C}\right)\;=\;\frac{{\bar{C}}^{\;q}-\bar{C}}{1-q}.$$The q→1 limit of this formula is the Boltzmann–Gibbs–Shannon relation:(16)$$\sigma_{1}\left(\bar{C}\right)\;=\;-\bar{C}\ln\bar{C}.$$The Gini index in the Tsallis–Pareto case is easily obtained as being
(17)$$G\;=\;\frac{2}{1-q}\int_{0}^{1}\;\left({\bar{C}}^{\;q}-\bar{C}\right)d\bar{C}\;=\;\frac{1}{q+1}.$$The formal analogy between the expressions of gintropy in terms of the tail-cumulative data population on the one hand and the entropy density in terms of the PDF on the other hand is obvious (cf. Equation (Equation 15)). Moreover, the general form of trace entropy is given as
(18)$$S\;=\;\int_{0}^{\infty }\;dx\;P\left(x\right)\;s\left(1/P(x)\right),$$
while the Gini index is obtained according to our previous discussion above as
(19)$$G\;=\;\int_{0}^{\infty }\;dx\;P\left(x\right)\;2\sigma \left(\bar{C}\left(x\right)\right).$$Here, we utilize the fact that
(20)$$\int_{0}^{1}d\bar{C}\;f\left(\bar{C}\right)\;=\;\int_{0}^{\infty }\;dx\;P\left(x\right)\;f\left(\bar{C}\left(x\right)\right)$$
for an arbitrary integrand, $f(\bar{C}\left(x\right))$.

Despite the intriguing analogies, we do not have a quantity that would be equivalent to the total entropy in social and econophysics. On the other hand, the nontrivial identification, $2\sigma \left(\bar{C}\left(x\right)\right)=s\left(P\left(x\right)\right)$, would make the Gini index equal to the entropy, G=S. Since P(x) is a negative derivative of the cumulative function $\bar{C}\left(x\right)$, the above G=S correspondence is a complex differential equation for $\bar{C}$. It may therefore be valid only for a single PDF, P(X), for the solution of the above implicit differential equation. In conclusion, gintropy cannot be replaced by entropy for a general PDF.

Let us review, briefly, how to obtain the general trace-form entropy once the gintropy, $\sigma (\bar{C})$, is known. To begin with, one uses a general function, s(1/P), in the definition of entropy with the required non-negativity and convexity properties. Due to its relation to the fiducial PDF, P(x), and using Equation (Equation 18), we obtain
(21)$$S\;=\;\int_{0}^{\infty }\;dx\;\;P\left(x\right)\;s\left(1/P(x)\right)\;=\;\int_{0}^{1}\;d\bar{C}\;\;s\left(-\langle x\rangle \sigma^{\prime \prime }\right)$$
with the short-hand notation
(22)$$\sigma^{\prime \prime }\;\equiv \;\frac{d^{2}\sigma }{d{\bar{C}}^{2}}.$$In particular, the Boltzmann entropy becomes
(23)$$S_{\mathrm{BG}}\;=\;\ln\langle x\rangle \;+\;\int_{0}^{1}\;d\bar{C}\;\ln\left(-\sigma^{\prime \prime }\left(\bar{C}\right)\right).$$

## 4. Dynamics of the Gini Index

After the introduction of gintropy, the authors of [32] provided several examples for different socioeconomic systems and compared the inequality measure *G* for their wealth distribution. Here, we supplement this steady picture with a dynamic one. We demonstrate, based on the example of the linear growth with reset (LGGR) model [33,34], that the Gini index mostly (i.e., not accounting for a short overshoot period, probably of numerical origin) increases monotonically, as the wealth distribution tends towards the stationary Tsallis–Pareto distribution. This behavior of the Gini index is not yet proven for the general case, in contrast to the entropy, cf. [32].

As in [32], the society members may have k≥0 discrete units of wealth. We assume that these members of the society acquire another unit of wealth with a rate that is linear to their actual wealth value (the rich get richer effect). We also incorporate a constant reset rate as in [32].

The evolution equation for the probability density function of the wealth distribution in the LGGR model is applied here to a binned wealth representation. In this case, the evolution equation, denoting $\frac{\partial P}{\partial t}$ with an overdot, reads
(24)$$\dot{P}\left(k,t\right)\;=\;\mu \left(k-1\right)P\left(k-1,t\right)-\left(\mu (k)+\gamma (k)\right)P\left(k,t\right),$$
where P(k,t) is the actual fraction of people in the wealth slot around *k*. In other words, one becomes richer with a state-dependent rate, μ(k), while there is a reset mechanism to zero wealth with the rate γ(k). This means not only a ruin probability rate, but also includes any type of exit of people, receiving the income *k*, from the studied population (e.g., resorting to pensions or the decay of hadrons containing energy *k*). The boundary condition at P(0,t) ensures that $\sum_{k=0}^{\infty }P\left(k,t\right)=1$ remains constant in time. This requirement results in
(25)$$\dot{P}\left(0,t\right)\;=\;\langle \gamma \rangle \left(t\right)\;-\;\left(\gamma (0)+\mu (0)\right)P\left(0,t\right),$$
with $\langle \gamma \rangle \left(t\right)=\sum_{k}kP\left(k,t\right)$.

We solve Equation (Equation 24) as a time recursion problem, with the linear μ(k)=ak+b and the constant γ(k)=γ parameter functions. In the numerical simulation, we discretize the possible values of *k* and use them as an integer index. Starting from a theoretical society where everybody has zero wealth, P(k,0)=δ(k) is represented by a Kronecker delta $\delta_{k,0}$, delivering a vanishing Gini index, G=0. Moreover, the whole Lorenz curve shrinks in this case to the diagonal and correspondingly the gintropy vanishes everywhere as a function of either *k* or $\bar{C}\left(k\right)$.

The growth rate μ(k), which is linear in *k*, is a common choice when dealing with the distribution of network hubs’ connection numbers and is called a preferential rate [35,36,37,38]. Obviously, the linear assumption is the mathematically simplest between all possible models. Nevertheless, further assumptions, such as a quadratic one, also can be made. The linear preference in the growth rate, utilized in the present discussion, together with a constant reset rate, has the Tsallis–Pareto distribution as the stationary PDF in the LGGR model.

We also observe in our numerical simulations that the Tsallis–Pareto power-law tailed wealth distribution develops, as was already anticipated in Ref. [32], cf. Figure 1. Furthermore, in Ref. [39], analytical expressions were given for the evolution of a general distribution for the cases with constant rates and for the presently discussed case of a linear growth rate with a constant reset rate.

We follow the time evolution of the Lorenz curve, $\bar{F}$ vs. $\bar{C}$, as well as the time-dependent Gini index. The results of the numerical calculation are shown in the upper and lower panels of Figure 2, respectively.

As can be observed, the wealth inequality grows in this theoretical example until it reaches its stationary position. The apparent slight overshoot at mid-time may be a numerical consequence of the time discretization. Recent, yet unpublished, analytical calculations of the time evolution of the Gini index in the very unique case studied numerically in the present paper indicate that G(t) would monotonically increase from zero to its stationary value. These somewhat laborious calculations will be published in a separate paper. On the other hand, since the Gini index is not an entropy underlying the second law in thermodynamics, the issue of the monotonity of the Gini index’s evolution in the general case calls for further investigations for a better understanding.

## 5. Summary

In summary, the quantity of gintropy, the difference between two tail-cumulative integrals of any PDF defined on non-negative values, features a formal dependence on the cumulative data population fraction having the form of various entropy formulas in terms of the original PDF [32]. In this paper, the particular form of Tsallis entropy was discussed in some detail.

The Gini index, used in economic studies to describe income and wealth inequality in societies, is an integral of the gintropy-cumulative data population fraction function. However, the Gini index–total entropy correspondence cannot be generally held, but only for a special PDF, given the trace entropy formula specification. Without this, the gintropic view of known entropy formulas can be obtained by expressing the PDF with the help of the gintropy’s second derivative with respect to the cumulative data population fraction and the average value of the base variable.

Time evolution in the particular but widespread case of a linear growth rate paired with a uniform reset rate was obtained numerically to demonstrate the evolution of the Gini index in time. A slight overshoot beyond its stationary value has been observed, so the Gini index does not appear to behave similarly to entropy in this particular case. However, to obtain a final conclusion, the scaling with the finite index space size should be studied.

## Figures and Tables

**Figure 1 entropy-26-00185-f001:**
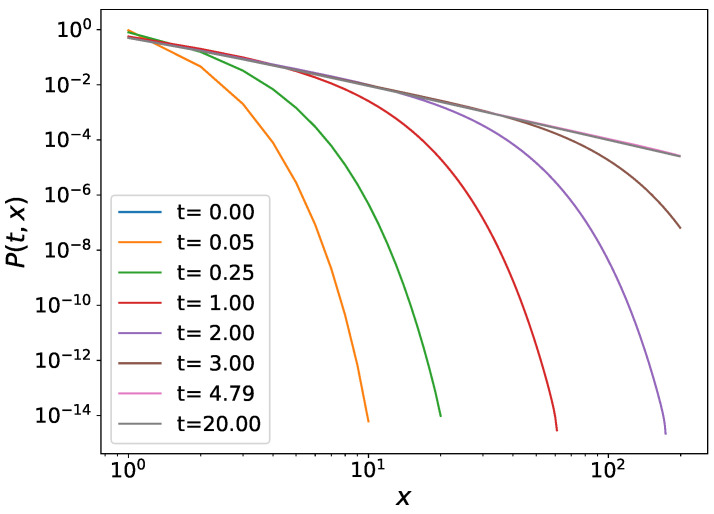
The time evolution of the wealth distribution starting from a society in which everybody has zero wealth.

**Figure 2 entropy-26-00185-f002:**
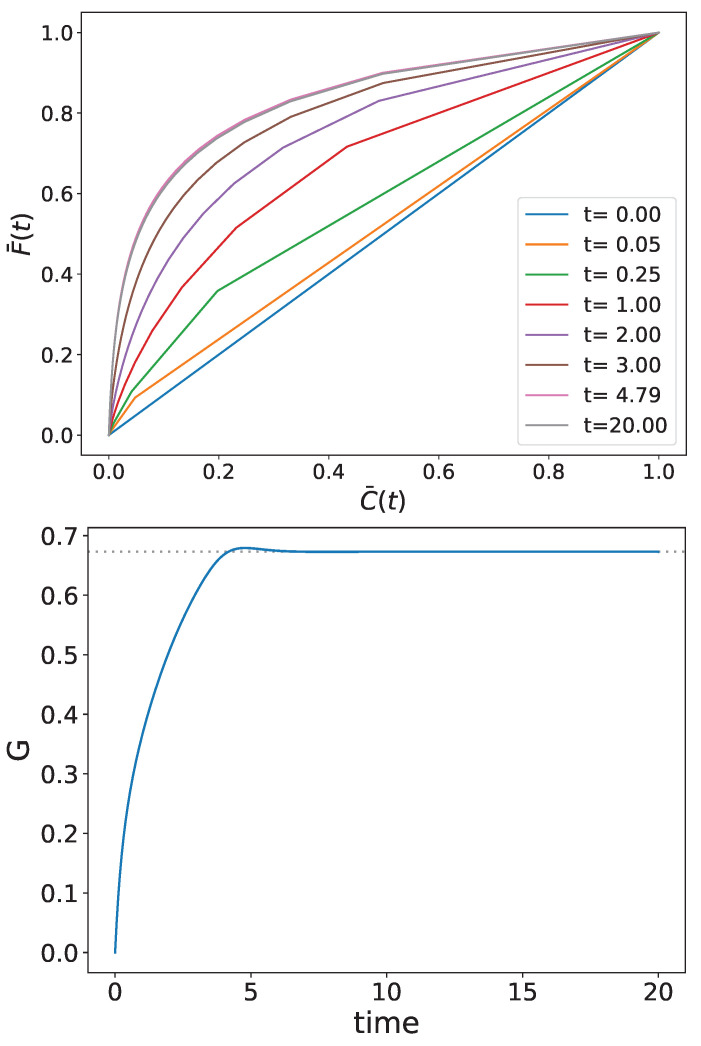
Time evolution of the Lorenz curve (**upper panel**) and the Gini index (**lower panel**). The steady dotted line in the lower panel corresponds to the final stationary Gini index.

## Data Availability

Data are contained within the article.

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
