# Peer review of "Analogies and Relations between Non-Additive Entropy Formulas and Gintropy"

_entropy, 2024, doi:10.3390/e26030185_

Round 1

Reviewer 1 Report

Comments and Suggestions for Authors

The article by Biró et al., in essence, represents a review of the past work of the authors on the notion of "Gintropy", complemented with some original aspects elucidated in the text, for instance, the discussion of the relation with Tsallis entropy. 

In my opinion, the manuscript is well written and is of help to the readers which aim at deepening into this interesting topic. 

I suggest the publication of the manuscript in its present form.

Comments on the Quality of English Language

I just wish to indicate a misprint: 

tis- > this in line 104.

Author Response

Thank you for reviewing the manuscript. We have corrected the typo. Further changes are due to inquieries appearing in other referee reports.

Reviewer 2 Report

Comments and Suggestions for Authors

attached

Comments on the Quality of English Language

attached

Author Response

Thank you for your reading and suggestions. Some of them we have implemented in our revised manuscript. Details:

  1. After considering your critique we have decided not to change the title.
  2. Indeed x is a general variable and the described methods are also more general than analyzing income data. Thank you for suggesting to apply it to parton distributions: once a PDF is obtained the Gini index and the dintropy can be calculated. However, such applications we would like to render to future papers.
  3. Differences between extensive and non-extensive temperatures reflect the different functional forms of the respective entropies used there. We do not think that a discussion of this issue is relevant for our analyses presented here.
  4. The suggested reference abozóut a general method obtaining non-extensivity from data is referenced now in the Introduction.
  5. Which entropy is meant? Any trace-form, with the definig densty function of 1/P satisfying the usual conditions. Now we elaborated on this around its defining equation.
  6. It is friendly to suggest to cite more of the previous works of the present authors. However, the mdpi editor is already against of about 16% self-citatuions. So we omit this possibility.

We hope that the revised manuscript meets the criteria of the high scientific standards revealed by the referee report.

Reviewer 3 Report

Comments and Suggestions for Authors

The submitted manuscript by Biró, Telcs, and Jakovác discusses, as the title indicates, relations between non-additive entropy formulas and the gintropy concept introduced by the authors earlier. The topic is quite relevant these days in high-energy phsyics, but probably in other areas of science as well. The paper first discusses non-additive entropy, then introduces gintropy. Subsequently these two concepts are connected, also to the so-called Gini-index. The paper is in general well written, and the calculations are mostly clear in terms of their mathematical details. The only major problem with the paper is that some of the explanations are not detailed enough, often there is just a small sentence connecting two statements with a "since" or "hence", but the flow of logic is not explained. Some examples are discussed below - in particular those about the logic and the reasoning should be responded to in detail.

L30 and other places: space in eq.(30) is missing, and some other places as well

L33: "trace form entropies" -> "trace-form entropies"

L33: "the uniform distribution" -> in what sense is this uniform? All states shall have an equal probability? It is strange to write this, as with a fixed total energy (in the microcanonical approach) this is usually not true.

L37: "inverse function" -> in what sense is this an inverse?

L39: "q-entropy" -> "$q$-entropy", if it is the variable (elsewhere as well)

L42: "tail cumulative fractions" -> "tail-cumulative fractions"? Although this does not seem like an established naming convention, so it would be better to define it (or write it as "tail of the cumulative distribution" or something similar)

L43: "and the definition" -> "We furthermore review" (and end the previous line with a full stop - it would be more clear this way)

L45: What does "dubbings" mean here?

Line before Eq (4): <x+y>=2<x> is for example true if x and y are random variables with the same probability density distribution, but this is not stated anywhere

L50: Space in "Ref.[28]" is missing (elsewhere as well)

Eq. (6): Wouldn't <y> be more clear here instead of <x> if they are the same?

L54: "data" -> "random variable"?

A few lines before Eq. (7): "above the diagonal" -> I guess it may also run on the diagonal (for a uniform distribution)

Second line of Sec. 3: "leads to vanishing gintropy" -> maybe this could be illustrated, along with some other simple examples, or a concrete plot of Ref. [28] be quoted?

L70: It is entirely not clear why "gintropy cannot be replaced by entropy", or a few steps from the reasoning are missing in the above text.

L75: A reference for the LGGR model would be useful here.

Eq. (24): dot P(k,t) is probably equal to partial P(k,t) / partial t, but this should be defined.

2-3 lines after Eq. (24): why does the last term lead to a reset rate of gamma(k), and what exactly does reset rate mean here? Maybe it is worth explaing this a bit more.

L85: What is the motivation for mu(k)=ak+b?

L87: delta_k0 is probably delta(0), the (Schwartz) distribution called Dirac-delta - the notation with "k0" in the index is a bit confusing, since k0 as a quantity was never introduced, and it is anyway centered around zero.

Fig. 1: Labels and legends are way too small - they should have a font size at least equal to the captions.

Fig. 1: Also, would it be useful to plot a Tsallis distribution on this plot:

Fig. 2: Again labels and legends are way too small.

Caption of Fig. 2: "dotted line" -> I don't see any line being dotted, so I would suggest that line widths are also increased.

L96: "may be" -> was this somehow tested?

L97: "since" -> what is the reason behind this causation?

L103: "reminding to" -> strange wording, maybe "similar to"?

L107: "hold" -> "held"?

Summary section: The conclusions are not really clear, in particular it could be explaned better why exactly the statements hold.

References: The list should be unified in terms of abbreviating first names or writing them out.

Comments on the Quality of English Language

(Language comments are intertwined with content-related ones above.)

Author Response

Thank you for your numerous suggestions to imporve the presentation of our concepts and results in this paper. We have followed all of your suggestions in the revised manuscript. In some cases it has been done by a minor modification of formulations in the text, in other cases correcting typos.

  1. Missing spaces, the spelling of trace-form and similar expressions are now following your suggestions.
  2. uniform distribution -> distribution uniform in x.
  3. inverse function: the contraint is linear in the energy butv not in the PDF. Therefore the entropy - PDF relation must be inverted. The inverse function phrase is used in the mathematical sense of undoing certain mappings.
  4. By q-entropy we are using the $q$-entropy form now.
  5. The expression tail cumulative is changed to tail-integrated cumulative.
  6. We applied your suggestion to break the long sentence in L43.
  7. The phrase dubbings is changed to "doubles", as it was meant so.
  8. We have inserted stating that <x+y>=2<x> when taken from tzhe same distribution.
  9. Space in Refenece citations are now inserted.
  10. Since <y>=<x> one may use them interchangebly. We prefer tzhe form <x> to be clear that overall same quantity is meant.
  11. above the diagonal -> above and on the diagonal
  12. The vanishing gintropy case is now explained by an additional half sentence.
  13. We increased the explaining text why gintropy in general cannot be replaced by entropy. They are desribing different quantities, even if featuring analogous forms.
  14. Two new citations of the LGGR model are inserted.
  15. dot{P} is explained now as partial time derivative by its first occurence.
  16. We have commented the reset and growth rates and their particular choices at various places in the revised manuscript.
  17. The motivation for a linear prefernce in the growth rate is now described in more details and 4 new refernces to network studies are inserted.
  18. delta(k,0) is inserted now, meaning the Kronecker delta. We elaborated on the explanation of this notation.
  19. Labels and figures are now enlarged, the refrence line is now really dotted.
  20. We have checked that the stationary distribution is indeed a Tsallis-Pareto one and at t=20 teh actual PDF coincides with it within line thickness.
  21. May be and since: the whole paragraph has been reformulated.
  22. reminding to -> similar to, cirrected.
  23. hold -> held corrected.
  24. We elaborated on the conlcusion text.
  25. By some of the references we do not know the full names, so we have changed all citations containing initials.

Round 2

Reviewer 3 Report

Comments and Suggestions for Authors

The authors responded to the review comments in an acceptable manner, so the paper can now be accepted.

Comments on the Quality of English Language

Paper reads well now.